# Exploring the Aroma Fingerprint of Various Chinese Pear Cultivars through Qualitative and Quantitative Analysis of Volatile Compounds Using HS-SPME and GC×GC-TOFMS

**DOI:** 10.3390/molecules28124794

**Published:** 2023-06-15

**Authors:** Wenjun Zhang, Mengmeng Yan, Xinxin Zheng, Zilei Chen, Huidong Li, Jiangsheng Mao, Hongwei Qin, Chao Zhu, Hongxia Du, A. M. Abd El-Aty

**Affiliations:** 1Institute of Quality Standard and Testing Technology for Agro-Products, Shandong Academy of Agricultural Sciences, Ji’nan 250100, China; zipingguozhang@163.com (W.Z.); ynky202@163.com (M.Y.); czl7274@163.com (Z.C.); lihuidong8066@163.com (H.L.); maojiangsheng@163.com (J.M.); qhw01@163.com (H.Q.); ndytzhuchao@126.com (C.Z.); 2Shandong Provincial Key Laboratory Test Technology on Food Quality and Safety, Ji’nan 250100, China; 3School of Bioengineering, Qilu University of Technology (Shandong Academy of Sciences), Ji’nan 250353, China; m17861407185@163.com; 4Department of Pharmacology, Faculty of Veterinary Medicine, Cairo University, Giza 12211, Egypt; abdelaty44@hotmail.com; 5Department of Medical Pharmacology, Medical Faculty, Ataturk University, Erzurum 25240, Turkey

**Keywords:** *Pyrus ussuriensis* Maxim, aroma characteristics, aroma profiles, GC×GC-TOFMS, HS-SPME, quality evaluation

## Abstract

To comprehensively understand the volatile compounds and assess the aroma profiles of different types of *Pyrus ussuriensis* Maxim. Anli, Dongmili, Huagai, Jianbali, Jingbaili, Jinxiangshui, and Nanguoli were detected via headspace solid phase microextraction (HS-SPME) coupled with two-dimensional gas chromatography/time-of-flight mass spectrometry (GC×GC-TOFMS). The aroma composition, total aroma content, proportion and number of different aroma types, and the relative quantities of each compound were analyzed and evaluated. The results showed that 174 volatile aroma compounds were detected in various cultivars, mainly including esters, alcohols, aldehydes, and alkenes: Jinxiangshui had the highest total aroma content at 2825.59 ng/g; and Nanguoli had the highest number of aroma species detected at 108. The aroma composition and content varied among pear varieties, and the pears could be divided into three groups based on principal component analysis. Twenty-four kinds of aroma scents were detected; among them, fruit and aliphatic were the main fragrance types. The proportions of aroma types also varied among different varieties, visually and quantitatively displaying changes of the whole aroma of the different varieties of pears brought by the changes in aroma composition. This study contributes to further research on volatile compound analysis, and provides useful data for the improvement of fruit sensory quality and breeding work.

## 1. Introduction

*Pyrus ussuriensis* Maxim is a pear species native to northeastern China that is known for its cold and drought resistance and the ability to grow in barren areas [1]. It is the primary pear cultivated in cold regions of China, with oblate or nearly round fruits that can be green, yellow, or yellowish-green, and red on the sunny side. Nanguoli, the most commonly grown cultivar, covers approximately 72,000 hectares of land and yields approximately 400,000 tons in China [2]. The fruits of *Pyrus ussuriensis* Maxim typically require postripening to develop a rich and excellent flavor [3]. Varieties such as Nanguoli and Jianbali are also suitable for producing high-quality frozen pears with good sour and sweet flavors [4]. In addition to its distinctive quality [5], *Pyrus ussuriensis* Maxim has various beneficial properties, such as acaricidal, antioxidant, and anti-inflammatory effects [6,7,8], which can help alleviate cough, gout, and rheumatoid arthritis. Researchers have found that *Pyrus ussuriensis* Maxim contains higher levels of polyphenols and triterpenic acids than other pear species [9], making it an excellent candidate for breeding purposes.

The scent of fruit is a crucial factor that affects the overall aroma and consumer preference [10]. It originates from the secondary metabolites of plants, which is an essential part of fruit quality and reflects the differences between species and varieties [11]. The analysis and evaluation of fruit aroma can aid in variety identification, quality assessment, breeding, and fruit processing [12,13,14,15]. Numerous studies have explored the composition of volatile aroma in pears and have shown that the aroma of pears is made up of esters, aldehydes, alkenes, alkanes, and other compounds [13]. Various factors, such as pollination, bagging, harvesting period, storage conditions, and calcium regulation, can impact pear aroma [16,17,18,19]. Regarding *Pyrus ussuriensis* Maxim aroma evaluation, some scholars have evaluated the types, quantity, and volatile content of the aroma in *Pyrus ussuriensis* Maxim, including Nanguoli, Huaguai, and Jianbali [13,20,21], mainly using headspace solid-phase microextraction (HS-SPME) in combination with gas chromatography-mass spectrometry (GC-MS) technology. Some researchers have also studied the physiological mechanism of aroma formation and release in “Nanguoli” fruit and identified relevant candidate genes via calcium regulation, providing valuable data for further research into the molecular mechanism of pear fruit aroma regulation [19]. However, the number of aroma components reported in *Pyrus ussuriensis* Maxim is still limited, and the analysis of aroma characteristics has not been adequately reported.

The comprehensive two-dimensional gas chromatography/time-of-flight mass spectrometry (GC×GC-TOFMS) technique was developed in the 1990s to identify volatile substances in a qualitative and quantitative manner. Compared to gas chromatography, GC×GC-TOFMS offers several advantages, including detecting more compounds, having a prominent peak capacity, good sensitivity, high resolution, and fast analysis [22]. Compared to QMS, TOFMS is especially useful in identifying volatile and semi-volatile compounds [23] and providing a comprehensive volatile fingerprint with its higher data acquisition frequency. As a result of these benefits, GC×GC-TOFMS has been widely used in the analysis of wine [24], cloud water [25], and green tea [26]. However, it has not yet been used to analyze *Pyrus ussuriensis* Maxim. Therefore, in order to comprehensively understand and evaluate the aroma quality of *Pyrus ussuriensis* Maxim, this study adopts the HS-SPME combined with GC×GC-TOFMS to demonstrate aroma fingerprint information of Anli, Dongmili, Huagai, Jianbali, Jingbaili, Jinxiangshui, Nanguoli. This study will characterize comprehensively volatile compounds, and provide more information on *Pyrus ussuriensis* Maxim aroma fingerprint. A comprehensive evaluation will be performed for each variety for the aspect of the concentration and number of volatile substances, volatile aroma biosynthetic, and aroma characteristic percentage distributions. Ultimately, we expect to provide valuable data for promoting pear fruit aroma quality, deep processing, and genetic breeding.

## 2. Results

### 2.1. Analysis of Aroma Composition in Different Pear Fruits

The GC×GC-TOFMS total ion flow diagram of aroma components of different cultivars pears is shown in Figure 1. After retrieving and analyzing data from the NIST/WILEY spectrum database, the main aroma components of pears were identified and are listed in Appendix A. A total of 174 volatile organic compounds were identified in the seven pear cultivars, comprising 102 esters, 14 alcohols, 20 aldehydes, 5 ketones, 18 alkenes, 8 alkanes, and 7 other substances. Hexyl acetate, ethyl acetate, butyl acetate, ethyl caproate, methyl caproate, methyl acetate, ethyl butyrate, ethyl caprylate, ethyl 2-methyl butyrate, octyl acetate, (*Z*)-3-hexenyl acetate, heptyl acetate, pentyl acetate, ethyl phenylacetate, 1-hexanol, 1-octanol, (*Z*,*E*)-*α*-farnesene, and hexanal were detected in all seven pear cultivars, in relatively high contents. Among the pear cultivars, Jinxiangshui had the highest total content of aroma substances at 2825.59 ng/g, which was 1.5–5 times higher than those of the other six pear cultivars. Anli (1877.71 ng/g), Dongmili (1799.46 ng/g), and Jianbali (1158.07 ng/g) had higher total contents of aroma substances than Nanguoli (657.67 ng/g), Huagai (608.53 ng/g), and Jingbaili (568.84 ng/g). Nanguoli had the largest number of aroma species (108), followed by Jiangubali (93) and Huagai (91), while Dongmili, Jinxiangshui, and Anli had the smallest numbers of aroma species (66, 64, and 64, respectively).

Esters were found to be the main aroma substances in the seven types of pears by comparing the number and relative percentages of different aroma types. The number of esters identified ranged from 40 to 72 (as shown in Figure 2), with the relative content percentage accounting for 56.68–94.92% (as shown in Figure 3). Although only 48 types of esters were identified in Jinxiangshui, its content was the highest, and the total amount of aroma substances was also higher than the other six types of pears (Appendix A). This suggests that the relative content of primary esters in Jinxiangshui was generally higher than that in the other six types of pears, which was further supported by the clustering heatmaps (as shown in Figure 4). The first cluster in Figure 4 shows that hexyl acetate, ethyl acetate, ethyl butyrate, ethyl caproate, methyl caproate, methyl acetate, and ethyl butyrate in Jinxiangshui were significantly higher than in the other six types of pears. These aroma components were also the main ester substances in all pear cultivars, as shown in Appendix A.

The relative percentages of esters in Anli, Dongmili, Huagai, and Nanguoli ranged from 85% to 91%, while in Jianbali and Jingbaili, these were less than 80%. Dongmili had the second-highest percentage of esters (91.17%) after Jinxiangshui, and the number of ester species was 50. Butyl acetate, ethyl acetate, hexyl acetate, ethyl crotonate, methyl acetate, and heptyl acetate were the main ester components of Dongmili. Heptyl acetate and isobutyl acetate had the highest relative content in Dongmili, according to Figure 4 and Appendix A. Anli had a percentage of esters of 90.96%, and the number of species was the same as that of Jinxiangshui. Hexyl acetate, ethyl acetate, butyl acetate, ethyl caproate, methyl caproate, methyl crotonate, methyl acetate, and ethyl butyrate were the main ester components of Anli. The relative content of hexyl acetate was significantly higher in Anli than in the other six types of pears (Figure 4 and Appendix A). Huagai had a percentage of esters of 87.80%, and the number of species was the same as that of Dongmili. The primary esters were hexyl acetate, butyl acetate, and (*Z*)-hex-2-enyl acetate, with the highest relative content of (*Z*)-hex-2-enyl acetate in Huagai, as shown in Figure 4 and Appendix A. Nanguoli had the largest number of ester species at 72, but the relative proportion was 85.87%. The main ester components were ethyl acetate, butyl acetate, methyl acetate, methyl butyrate, ethyl caprylate, ethyl 2-methyl butyrate, hex-2-enyl acetate, and methyl octanoate. The relative contents of ethyl octylate and methyl octylate were the highest in Nanguoli, as shown in Figure 4 and Appendix A. Jianbali had a lower proportion of esters (77.42%) and contained 59 esters. The main ester components were hexyl acetate, ethyl acetate, isoamyl acetate, ethyl caproate, methyl acetate, butyl acetate, and propyl acetate. Isoamyl acetate was only detected in Jianbali, as shown in Appendix A. Jingbaili had the lowest proportion of esters (56.68%) and the lowest number of species (Figure 3).

The percentage of alcohols varied between 3.43% and 21.94%, and the number of species ranged from 4 to 10 (Figure 2 and Figure 3). Ethanol, 1-hexanol, and 1-octanol were present in all seven types of pears and had higher relative contents than other alcohols (Appendix A). Jingbaili had the highest alcohol percentage (21.94%); the relative contents of (*E*)-2-hexanol were higher than those of the other six types of pears, and 4-methyl-1-pentanol (26.08 ng/g) was only detected in Jingbaili (Appendix A). The second was Jianbali, accounting for 11.41%, of which (±)-2-methyl-1-butanol was only detected (Appendix A). Nanguoli, Huaguai, and Dongmili had proportions of alcohols ranging from 7.54% to 5.06%. Only Nanguoli contained 1-nonanol (Appendix A), and only Huagai had 3,7,11-trimethyl-6,10-dodecadiene-1-ol and 3-octanol (Appendix A). The proportion of alcohols in Anli and Jinxiangshui were all below 4%. (*Z*)-2-hexene-1-ol was only present in these two pear species, with a higher content in Jinxiangshui (5.52 ng/g) (Appendix A and Figure 4).

In the seven types of pears, aldehydes were also found to be critical volatile compounds, accounting for 1.08%–7.56%, and the number of detected species ranged from 4 to 14 (Figure 2 and Figure 3). The most abundant aldehydes detected were hexaldehyde and (*E*)-2-hexenal. Jingbaili had the highest proportion of aldehydes (7.56%) and the largest number of species (14) (Figure 2 and Figure 3). Croton aldehyde was found only in Jingbaili (Appendix A). Huagaili (5.45%) and Jianbali (5.11%) were second, with 10 and 13 aldehyde species, respectively (Figure 2 and Figure 3). (*Z*)-3-hexenal and (*E*)-2-heptenal were only detected in Jianbali (Appendix A). Dongmili had a lower proportion of aldehydes (3.35%), with only five species identified (Figure 2 and Figure 3). Anli (1.08%), Nanguoli (1.28%), and Jinxiangshui (1.31%) had low aldehyde percentages with 4, 10, and 4 species, respectively (Figure 2 and Figure 3). Although there were many aldehydes in Nanguoli, the content was low, among which 2-methylbut-2-enal and 2-methyl propenal were unique aldehydes in Nanguoli.

Alkenes accounted for 0.18–12.98% in the different types of pears (Figure 3), and the number of alkenes detected varied from 3 to 10 (Figure 2). Among the alkenes, (*Z*,*E*)-*α*-farnesene and *α*-farnesene were found in all seven pear types with higher contents. Additionally, (*Z*,*Z*)-*α*-farnesene and (*E*)-*β*-farnesene, other isomers of the above two substances, were also identified. (*Z*,*Z*)-*α*-farnesene was only detected in Nanguoli (Appendix A), and (*E*)-*β*-farnesene was detected in the five other pear species except for Dongmili and Jinxiangshui and had the highest content in Anli (2.91 ng/g). The highest percentage of alkenes (12.98%) was in Jingbaili, which also had the highest number of alkenes (10) (Figure 2 and Figure 3). Anli (4.09%) and Jianbali (4.74%) had alkenes in proportions above 4% (Figure 3), and their respective numbers of alkenes were 4 and 7. Dongmili (0.23%), Huagai (1.04%), Jinxiangshui (0.18%), and Nanguoli (3.35%) all had alkenes below 4%, with the number of species ranging from 3 to 9 (Figure 2 and Figure 3). Similar to the aldehydes, Nanguoli had a large number of alkenes but a low relative content (Figure 2 and Appendix A). Furthermore, some alkenes, such as 3-(4-methylpent-3-enyl)furan, (*Z*)-1,3-pentadiene, (*E*)-1,4-hexadiene, (*E*)-3-hexene, and (*E*)-1-methyl-4-(6-methylhept-5-en-2-ylidene) cyclohex-1-ene, were only detected in Jingbaili, Jianbali, and Huagai (Appendix A). Additionally, *α*-curcumene was only found in Jinxiangshui (Appendix A). Finally, this study identified three monoterpenes, limonene, *β*-myrcene, and (*E*)-*β*-ocimene, and although their concentration was low, they still played an important role in fruit flavor [27]. Limonene was only present in Huagai (0.04 ng/g) and Nanguoli (0.19 ng/g), while *β*-myrcene (0.04 ng/g) and (*E*)-*β*-ocimene (0.03 ng/g) were only detected in Nanguoli (Appendix A).

The content and species of ketones in seven types of *Pyrus ussuriensis* Maxim were lower; the percentage of ketones ranged from 0.01% to 0.34%. The higher proportions of ketones were found in Jingbaili (0.34%) and Nanguoli (0.30%), which accounted for over 0.3% (Figure 3). Jingbaili had both methyl heptenone and butenone, while methyl heptenone, a crucial volatile component of Fuzhuan tea [28], was detected in all types except Dongmili. Butenone was only present in Jingbaili and Nanguoli. Moreover, Nanguoli had (*E*)-6,10-dimethyl-5,9-undecadien-2-one, which is commonly found in tobacco and tea and has a distinct floral and woody fragrance [29]. The proportions of ketones in Huagai (0.11%) and Jianbali (0.19%) were all above 0.1% (Figure 3), and the number of ketones in pear species was 3 and 2, respectively (Figure 2). Huagai contained methyl heptenone, (*E*)-6,10-dimethyl-5,9-undecadien-2-one, and 2,3-dioctyl ketone, the latter of which was only detected in Huagai. The ketone substances in Jianbali were methyl heptenone and 2,3-butanedione, with the latter being exclusively found in Jianbali (Appendix A). The percentages of ketones in Anli (0.06%) and Jinxiangshui (0.01%) were relatively low, accounting for less than 0.1% (Figure 3), and only methyl heptenone was detected (Appendix A).

The content of alkanes in seven types of *Pyrus ussuriensis* Maxim was lower, and the proportion ranged from 0.03% to 0.39% (Figure 3). Except for Anli, different types of alkanes were detected in the six other pears (Appendix A). Only *o*-xylene was identified in Dongmili, which has a distinctive aromatic odor [30]. Both Jianbali and Nanguoli contained *p*-xylene, a compound with a sweet and fruity aroma [31]. Denderalasin, which has not been previously reported in pear aroma identification, was the only compound belonging to alkanes detected in Huagai.

### 2.2. Principal Component Analysis

A total of 174 volatile components present in seven types of *Pyrus ussuriensis* Maxim were analyzed by SIMCA14.1 software. The variance contribution rate, cumulative variance contribution rate and characteristic value of each principal component are shown in Table 1. The results indicate that the six principal components, which had eigenvalues greater than 1 and an accumulative variance contribution rate of 97.0%, reflect all the information of the original variable. Based on the identified aroma substances and their relative content in different types of pears, the seven types were categorized into three groups (Figure 5). The gray spheres in Figure 5 represent the aroma substances, and the closer they are to the hexagon representing the samples, the greater the contribution to sample classification. Appendix A indicates the primary substances contributing to the grouping. Group 1 comprises Jianbali and Jingbaili, which were clustered in the first quadrant, and 2-methylbutyl acetate(*E*)-2-hexenoic acid methyl ester, diethyl carbonate, (*E*)-2-hexen-1-ol, ethyl tiglate, and other substances played a crucial role in the grouping. Nanguoli was the only pear type in Group 2, clustered in the third quadrant with some C_6_ and C_9_ compounds, such as 2-hexenal, (*E*)-2-nonenal, 1-nonanol, some unsaturated esters, alkenes, sulfur-containing ester compounds, hexyl ester compounds, and ethyl compounds, such as (*Z*)-4-decenoic acid methyl ester, limonene, *β*-myrcene, (*E*)-*β*-ocimene, (*Z*,*Z*)-*α*-farnesene, 3-(Methylthio)propanoic acid ethyl ester, hexyl caprylate, and ethyl heptanoate, which significantly affected the separation between Nanguoli and other pear cultivars. Group 3 comprises Anli, Huagai, Dongmili, and Jinxiangshui, clustered in the fourth quadrant, characterized by high concentrations of hexyl acetate, ethyl (*2Z*)-but-2-enoate, (*Z*)-hex-2-enyl acetate, ethyl (*E*)-hex-3-enoate, methyl tiglate, 1-octanol, *n*-heptanol, and other substances.

### 2.3. Analysis of Aroma Characteristic Percentage Distributions

In seven varieties of pears, a total of 24 types of flavors were identified, with 18 common flavors including aliphatic, ice, citrus, dairy, edible, fruit, green, herb, konifer, it-chem, narcotic, orchid, phenol, rose, animal, wood, earthy, and solvent flavors. Anli, Dongmili, and Jinxiangshui exhibited a higher total distribution values of all aroma characteristics, with Jianbali following closely behind, and the remaining pear varieties had lower values (Appendix A). Among all the flavors, fruit aroma, accounting for a significant portion of the total flavor profile (44.12–82.90%), was the most prominent aroma of the seven pear varieties compared with aliphatic aroma (2.79–19.46%), green aroma (5.18–16.76%), dairy aroma (1.81–5.22%), and solvent aroma (1.25–5.67%). Anli had the highest proportion of fruit flavor, followed by Jinxiangshui (76.90%), Huagai (75.94%), Jianbali (71.31%), Dongmili (66.10%), Nanguoli (62.45%), and Jingbaili (44.12%). The main contributors to fruit flavor were short-chain saturated esters, particularly ethyl ester compounds such as hexyl acetate, ethyl acetate, isoamyl acetate, and butyl acetate (Appendix A), with the percentage of relative contents of approximately 77.87% and 76.05% in Jinxiangshui and Anli, respectively. The second most abundant aroma types were aliphatic and green. Jingbaili had the highest aliphatic aroma percentage (19.46%), followed by Dongmili (17.38%) and Nanguoli (11.43%), while it was below 10% in the other pear cultivars (Figure 6). Dongmili had a high proportion (75.99%) of saturated esters. However, it also contained a higher proportion of saturated alcohols (5.06%) and aldehydes (3.35%), which significantly contributed to aliphatic and other scents, compared to Jinxinagshui (3.43%, 1.31%) and Anli (3.69%, 1.08%) (Appendix A). Therefore, this could reduce the proportion of fruit flavor and the “intensity” of the fruit aroma in Dongmili to some extent. Comparative analysis also revealed that although the total value of aliphatic aroma in Dongmili was significantly larger than that in Jingbaili (Appendix A), the proportion of aliphatic aroma in Jingbaili was larger than that in Dongmili. This is because the total aroma value of Dongmili is large, which reduces the proportion of aliphatic aroma. These findings suggest that the combined presence of different aroma types has a significant impact on the overall aroma of pears. The green scent accounted for 5.18–16.76% (Figure 6), contributed by ethyl caproate, methyl caproate, methyl acetate, (*Z*)-3-hexenyl acetate, heptyl acetate, (*E*)-2-hexen-1-ol, hexanal, and some alkene substances. Jingbaili (16.76%) and Dongmili (10.58%) had a higher proportion of green aroma, which was more than 10% of the total aroma characteristics. The percentage of dairy aroma, accounting for 1.81–5.22%, was also significant, mainly contributed by isoamyl acetate, butyl acetate, ethyl caproate, and methyl acetate. Jinxiangshui (5.22%) and Jianbali (4.46%) had a higher proportion, and the remaining pear species had less than 4% dairy aroma. The percentage of solvent aroma ranged from 1.25% (Dongmili) to 5.67% (Nanguoli) (Figure 6) and was primarily produced by methyl acetate and other short-chain esters, such as *n*-hexanol. The presence of citrus aroma (as in many citrus fruits [32] and green tea [33]), accounting for 0.003% (Jinxiangshui)–0.25% (Jingbaili), was also detected and mainly emitted by (*E*,*Z*)-2,4-decadienoic acid, methyl ester, nonanal, heptanal, and decanal. Similarly, the rose aroma, contributed by phenethyl acetate, diisobutyl phthalate, and *n*-heptanol, accounted for 0.01% (Jinxiangshui)–0.27% (Dongmili). Narcotic flavors, which were reported in *P. communis* L. [34] and were found in seven *Pyrus ussuriensis* Maxim in this study, were present in high proportions in Jinxiangshui, Anli, and Jianbali, accounting for more than 0.5% and contributed by ethyl caproate, ethyl benzoate, methyl benzoate, benzeneacetaldehyde, and benzaldehyde. Conifers and woody aroma types, which are mostly emitted by terpenes [35], were also detected, with Nanguoli (0.55%) having the highest proportion of conifers aroma, and Jingbaili (3.01%), Huagai (2.99%), and Jianbali (1.36%) having the highest proportion of woody aroma.

To summarize, Nanguoli were characterized by a dominant fruit and aliphatic aromas, along with some green, dairy, and solvent fragrances. Jinxiangshui, Jiandaili, Huagai, and Anli had a fruit-forward aroma, with some green, aliphatic, dairy, and solvent fragrances. Huagai and Jianbali had traces of it-chen, woody, and solvent aromas. Jingbaili and Dongmili had a complex aroma profile, with a dominant fruit, aliphatic, and green aromas, and with some edible, it-chen, woody, and solvent flavors. The analysis of aroma substance content and aroma type proportions revealed that the aroma of pears was affected not only by the main contributing substances but also by the composition of different aroma types.

Analyzing the number of aroma types is also a crucial aspect of characterizing aroma profiles. This study revealed that Nanguoli had 24 fragrance types, except for jasmine and spice, and had more iris, smoke, and musk than the other six pear species. Iris has been reported in *Pyrus communis* L. [17], and *β*-myrcene, a monoterpene found in Nanguoli and not presenting in the other pear cultivars (Appendix A), is responsible for the iris, burnt tobacco, and musk scents. This explains why Nanguoli has a more diverse flavor profile than other pear species to some extent. Moreover, some fragrances were only detected in a few pear species. For instance, the muguet scent, found in Yuluxiangli [17], was detected in Anli, Dongmili, Huagai, Jianbali, and Nanguoli in this study, primarily contributed by ethyl benzoate and diisobutyl phthalate. Balsam scent, detected in mango [36], was found only in Dongmili (0.16%), Huagai (0.0009%), and Nanguoli (0.004%), primarily contributed by diisobutyl phthalate. Vanilla scent, detected in *Pyrus communis* L. and Yuluxiangli [17], mainly contributed by ethyl phenylacetate and (*E*)-2-hexen-1-ol, was found in all five other pear species, except Dongmili and Jinxiangshui.

### 2.4. Main Biosynthesis Pathway of Aroma Components

Linear esters, aldehydes, alcohols, and ketones, the main aroma substances of seven pear cultivars, are derived from fatty acids and included in [16]. Short-chain acetate and medium-chain esters (C_6_–C_12_), belonging to linear esters, have relatively high contents in the seven pear varieties, including hexyl acetate (3.05–376.8 ng/g), ethyl acetate (7.14–391.96 ng/g), ethyl caproate (1.72–264.42 ng/g), methyl caproate (9.93–246.00 ng/g), ethyl butyrate (0.22–171.90 ng/g), methyl butyrate (7.88–129.01 ng/g), ethyl caprylate (0.35–42.40 ng/g), and methyl octanoate (4.90–32.96 ng/g) (Appendix A). Some of the above compounds give a strong “pear-like” flavor [37]. Volatile substances such as C_6_ and C_9_ alcohols, C_6_ and C_9_ aldehydes, have a “green grass” flavor, and are also the intermediate products of the fatty acid oxidation pathway. In this study, the seven pear varieties contained high levels of 1-hexanol (11.40–61.26 ng/g) and hexanal (3.03–65.40 ng/g). 1-nonanol and 2-nonanol were also identified in all seven *Pyrus ussuriensis* Maxim varieties, and nonanal was found in Anli (0.63 ng/g), Huagai (0.12 ng/g), Jianbali (0.52 ng/g), and Jingbaili (0.53 ng/g) (Appendix A).

Amino acid-derived compounds include branched-chain compounds, sulfur-containing compounds, and aromatic compounds [17]. In this study, various forms of branched-chain esters, such as 3-methyl-3-buten-1-ol, acetate, propanoic acid, 2-methyl-, 3-hydroxyl-2, 2,4-trimethyl pentyl ester, were detected at concentrations ranging from 0.34 to 1.27 ng/g (Appendix A), and ethyl 2-methyl butyrate (36.19 ng/g) had a relatively high content in Jianbali (Appendix A). The sulfur-containing ester compounds, such as ethyl 3-(methylthio)-(*E*)-2-propenoate, methyl 3-(methylthio)-(*E*)-2-propenoate-3-(methylthio), and methyl 3-methylthiopropionate, derived from methionine and cysteine [34], were only detected in Huagai and Nanguoli, with contents ranging from 0.21 to 0.68 ng/g. Diisobutyl phthalate, the only one aromatic ester, was detected in Huagai (0.26 ng/g) and Nanguoli (1.03 ng/g).

Carbohydrate-derived volatile substances are also an important group of compounds that can impact the flavor of pears to some extent [38]. Monoterpenoids and sesquiterpenoids were the main alkenes detected in the seven pear types. *β*-myrcene and (*E*)-*β*-ocimene, two monoterpene compounds, were identified in Nanguoli, with concentrations ranging from 0.03 ng/g to 0.04 ng/g. (*E*,*Z*)-2,4-decadienoic acid, methyl ester, and (*E*,*Z*)-2,4-decadienoic acid ethyl ester were esters composed of monoterpenoids. Among these, (*E*,*Z*)-2,4-decadienoic acid methyl ester was detected in all pear types except for Jingbaili, with concentrations varying from 0.08 ng/g (Anli) to 1.37 ng/g (Nanguoli). (*E*,*Z*)-2,4-decadienoic acid ethyl ester, was only identified in Anli, Jianbali, and Nanguoli, with concentrations ranging from 1.13 ng/g to 3.06 ng/g. Five sesquiterpenoids were also identified: *α*-farnesene, (*Z*,*E*)-*α*-farnesene, (*Z*,*Z*)-*α*-farnesene, (*E*)-*β*-farnesene, and limonene. *α*-farnesene and (*Z*,*E*)-*α*-farnesene were present in all seven pear types; *α*-farnesene had the highest content in Jingbaili (66.11 ng/g) and the lowest content in Dongmili (2.34 ng/g); (*Z*,*E*)-*α*-farnesene varied from 0.12 ng/g (Jinxiangshui) to 5.98 ng/g (Nanguoli). (*Z*,*Z*)-*α*-farnesene was only found in Nanguoli (0.25 ng/g). In addition to Dongmili and Jinxiangshui, the other five kinds of pear species all contained (*E*)-*β*-farnesene with contents from 0.14 ng/g to 2.91 ng/g, and limonene was detected in Huagai (0.04 ng/g) and Nanguoli (0.19 ng/g).

Based on the relative contents of the main aroma components in seven pear cultivars, the main aroma synthesis pathway of each pear can be preliminarily deduced. The fatty acid pathway was found to be the main pathway in all seven pear cultivars, with short-chain saturated esters making up over 50% of the aroma compounds in Anli, Dongmili, Huagai, Jianbali, Jingbaili, Jinshui, and Nanguoli. Other aroma biosynthesis pathways were also detected in the seven *Pyrus ussuriensis* Maxim samples besides the fatty acid synthesis pathway. For instance, the relative contents of amino acid derivatives in Huagai and Jianbali were over 3%, indicating the presence of the amino acid aroma biosynthesis pathway. In Jingbaili and Nanguoli, the relative contents of carbohydrate derivatives were 12.49% and 4.5%, respectively, while it accounted for a relatively lower proportion in the five other pear cultivars.

## 3. Discussion

A total of 174 volatile organic compounds were identified using HS-SPME in conjunction with GC×GC-TOFMS. Esters, primarily containing short-chain saturated acetates and medium-chain esters, were the dominant aroma components in both species in terms of quantity and proportion. Numerous researchers, including Gaihua Qin [13], Liping Zhang [20], and Jie Li [21], have investigated the aroma of *Pyrus ussuriensis* Maxim. They discovered that esters were the primary aroma compounds, which is consistent with our findings, but the total number of volatile organic compounds detected was only approximately 30 for Nanguoli, Huagai, Jingbaili, and Jianbali. Ethyl 3-methylthiopropionate was discovered by Takeoka in pineapple in 1989 [39] and was also detected in Nanguoli in this study. Furthermore, methyl 3-methylthiopropionate and ethyl 3-methylsulfonamido-propionate were found in Nanguoli. Other specialists found ethyl tiglate, an uncommon plant aroma compound [40], in Nanguoli. In this study, ethyl tiglate was found in all seven pear cultivars, propyl tigelate was found in Dongmili, and methyl tiglate was found in Huagai (Appendix A). Compared to other research, this study provides more detailed aroma fingerprint data, offering more extensive theoretical support for analyzing and assessing *Pyrus ussuriensis* Maxim germplasm resources and aroma breeding.

The flavor of pears is strongly influenced by the type and amount of aroma components present, which can be objectively evaluated using sensory tests. Numerous studies have employed sensory evaluation tests to investigate aroma properties in different types of fruits. For example, Hayaloglu et al. analyzed the phenolic compounds and sensory characteristics of 12 sweet cherry varieties [41], while Taiti et al. studied traditional and innovative Japanese plum cultivars grown in Italy [42]. Hernandez et al. explored the nutritional composition and sensory profile of Spanish jujube fruits [43]. Moreover, the critical flavor factors of fruits were investigated by integrating the substance content and sensory evaluation attributes. Sung et al. investigated the relationship between the sensory attributes and chemical components of different mango varieties [44]. Ramirez et al. used a quantitative descriptive-analytical method to investigate the sensory characteristics and factors affecting the freshness of seven watermelon varieties [45], while Siebert et al. studied the volatile compounds in Viognier and Chardonnay wines associated with “stone fruit” aroma properties [46]. Aroma characteristic analysis has been utilized in previous studies to assess fruit flavor attributes, such as Wang’s analysis of aroma types in *Pyrus communis* L. [17], and Zhang’s study of mango aroma attributes using fragrance characteristic analysis [36]. These studies have provided useful approaches for evaluating fruit flavor quality. This study employs aroma characteristic analysis to illustrate the flavor attributes of *Pyrus ussuriensis* Maxim. The study identifies fruit aroma, aliphatic aroma, and green aroma as the most important aroma types of the seven pears and quantitatively demonstrates the aroma attributes of different *Pyrus ussuriensis* Maxim. Based on the compound information, the main contributors to aroma types were identified. Through comparison, we found that the relative content of contributors to fruit aroma was basically the same, but the flavor characteristics were still different for Jinxiangshui, Anli, and Dongmili. Therefore, it can be concluded that the aroma characteristics were not only related to the content of the main contributors but also closely related to the proportion of aroma types in the total aroma value. Meanwhile, the change in aroma and the reason for aroma difference can be visually and quantitatively displayed according to the aroma characteristics of *Pyrus ussuriensis* Maxim. Thus, the aroma characteristics analysis of *Pyrus ussuriensis* Maxim provides a valuable tool for improving the aroma quality of this species and evaluating germplasm resources. This study also speculates on the major aroma biosynthesis pathways of the seven *Pyrus ussuriensis* Maxim, with the fatty acid oxidation pathway being the most crucial, in addition to the amino acid and carbohydrate biosynthesis pathways. The analysis of the aroma pathway provides some theoretical basis for the subsequent aroma synthesis pathway research on *Pyrus ussuriensis* Maxim.

## 4. Materials and Methods

### 4.1. Materials

For this study, seven cultivars of *Pyrus ussuriensis* Maxim were selected for analysis (Table 2 lists the cultivars, their production regions, and harvesting times). The maturity of the pears was determined using standard criteria such as days of growth, appearance, and seed color. All samples were kept refrigerated until treatment. A postripening process was necessary to enhance their flavor and taste. Samples were placed at room temperature before experiments (approximately five days) of this time. In addition, 50/30 μm DVB/CAR/PDMS was supplied by Supelco (Bellefonte, PA, USA). The length of the fiber coating was 1 cm. The internal standard 2-nonanone (>99%) was obtained from Dr. Ehrensorfer (Augsburg, Bavaria, Germany). A mixture of 500 μg/mL C_8_–C_40_ *n*-alkanes was purchased from Dr. Ehrenstorfer (Germany), and 2.0 mg/mL C_6_–C_10_ *n*-alkanes were obtained from AccuStandard (New Haven, CT, USA).

### 4.2. Methods

#### 4.2.1. Preparation of Standard Solution

First, 0.1 g of 2-nonanone (accurate to 0.0001 g) was diluted with 10 mL of methanol (HPLC grade, Thermo Fisher Scientific, Waltham, MA, USA) to prepare a 10,000 mg/L stock solution. From this, a 10 mg/L working solution of 2-nonanone was prepared using methanol (HPLC grade, Thermo Fisher Scientific, Waltham, MA, USA). Then, 500 μg/mL of C_8_–C_40_ *n*-alkane mix solutions and 2.0 mg/mL of C_6_–C_10_ *n*-alkanes were diluted to 10 mg/L with acetone (Analytical reagent, Sinopharm, China). All standard solutions were stored at −18 °C.

#### 4.2.2. GC×GC-TOFMS Conditions

An Agilent 7890B gas chromatograph (San Jose, California, USA) coupled with a LECO Pegasus 4D-C time-of-flight mass spectrometer (TOFMS) (Saint Joseph, CA, USA) was utilized for the analysis. Two columns were employed for compound separation, a weak polarity Rxi-5MS column (Shimadzu, Kyoto, Japan) (0.25 µm, 30 m × 250 µm) for the first dimension (1D) and a medium polar Rxi-17Sil MS column (Shimadzu, Kyoto, Japan) (0.25 µm, 2 m × 250 µm) for the second dimension (2D). The helium carrier gas (99.999%) was used as a carrier gas at a constant flow rate of 1.4 mL/min, while the split ratio was 10:1. The injection port and transmission line temperatures were 270 °C and 280 °C, respectively. The temperature program started at 40 °C for 2 min, heated to 200 °C at 5 °C/min, then rose at 20 °C/min to 280 °C and kept up for 2 min. During the entire collection, the second column temperature was maintained at 5 °C above the first column temperature, and the modulator temperature exceeded 15 °C compared with that of the second column. The modulation cycle was 3 s, and the thermal pulse time was 0.6 s.

The MS parameters were set as follows: the acquisition delay was 60 s, the acquisition rate was 100 spectra/s, the acquisition voltage was 1450 V, and the electron energy was set to −70 V. The ion source temperature was maintained at 250 °C. The mass spectra were collected over a range of 35–550 amu.

#### 4.2.3. Volatile Extraction

The extraction of isolated volatile organic compounds was performed using HS-SPME. A 50/30 µm DVB/CAR/PDMS SPME fiber was selected based on prior testing and published literature [27] and conditioned according to operational instructions before usage. Each pear sample was quartered following the quartic method, and the flesh without cores was cut into pieces measuring 0.5 cm × 0.5 cm × 0.5 cm. Subsequently, 6.0 g of the mixed sample was placed in a 15 mL headspace bottle, 5 μL of 10 mg/L 2-nonanone was added, and the vial was sealed, allowing it to stand for 10 min. The bottle was then kept at 40 °C for 40 min under thermostatic conditions, after which the SPME fiber was immediately placed into the GC injector for desorption. Each pear variety was tested three times in the experiment.

#### 4.2.4. Analysis of Aroma Characteristic Distributions

The aroma characteristics of seven pear varieties were evaluated using the “Training the ABCs of Perfumery” method [17]. In this method, all aroma compounds were categorized into 26 types and quantified using the “scent ABC”. The aroma characteristic percentage (Ac) was calculated using the following formula [36]:$$A_{c}=\frac{\sum_{i}^{n}\left(m_{i}\times I\times P_{i}\right)}{T}$$
where *A_c_* represents aroma characteristics (%); *m_i_* represents the relative amount of the aroma component; *I* represents the intensity of an aroma component compared to linalool with a fixed value of 100; *P_i_* represents the percentage of the odor of the substance in the 26 odors of A–Z (%); *n* is the number of aroma components with an ABC value; and *T* is the total distribution value of all aroma characteristics for each pear cultivar.

#### 4.2.5. Data Processing and Statistical Analysis

The NIST/WILEY.11 database was used to search for the mass spectrometry fragments of each chromatographic peak. Volatile aroma substances with matching degrees greater than 700 were selected and quantified using the internal standard method, with 2-nonanone as the standard substance for normalization. The following formula was used for the quantification of aroma substances [27]:$$C_{a}=\frac{\frac{PA_{a}}{PA_{is}\;}\times C_{is}\times 5\;\mu L}{m}$$
where *C_a_* is the concentration of aroma components (ng/g), *PA_a_* is the peak area of aroma components, *PA_is_* is the peak area of the internal standard, *C_is_* is the concentration of the internal standard (g/mL), and *m* is the mass of the sample (g). The concentration of 2-nonanone was 10 mg/L, and the mass of the sample was 6.0 g. The data presented are the means ± standard deviations of three replications.

The retention index was used to further confirm the accuracy of the selected aroma substances with degrees greater than 700. First, five microliters of 10 mg/L *n*-alkanes were analyzed according to step 2.2.3 to calculate the retention index of each aroma component. Then, the retention index data were confirmed with those searched from the NIST database. Finally, aroma substances were identified. The formula used to calculate the retention index of each aroma component was followed [27]:$$RI_{a}=\frac{RT_{a}-RT_{n}}{RT_{N}-RT_{n}}100\left(N-n\right)+100RT_{n}$$
where *RI_a_* is the retention index of the target compound, *a* is the target compound, *n* is the carbon number of the lower normal alkane, *N* is the carbon number of the higher normal alkane, and *RT* is the retention time.

SIMCA 14.1 was employed to conduct principal component analysis (PCA) and group pear varieties based on the identified volatile components. Tbtools was utilized to perform heatmap and clustering analysis (HCA) to classify all volatile organic compounds and represent their content through color depth. The data were analyzed using variance analysis in SPSS Statistics 19.0, and the mean values were compared using Tukey’s test with a significance level of *p* < 0.05. The results are presented as the means ± SDs with a sample size of 3.

## 5. Conclusions

The aroma of fruits is a crucial indicator of their quality and can reveal their flavor characteristics and ripeness. This study employed HS-SPME combined with GC×GC-TOFMS to increase the richness of aroma fingerprints in pear. The analysis identified a total of 174 chemical substances, including esters, alcohols, aldehydes, and alkenes, in seven different pear varieties. Based on this information, it was possible to cluster the pear varieties into groups with similar characteristic substances. Through the analysis of aroma characteristics, this study visually demonstrated how changes in the composition of aroma materials could affect the overall aroma quality of the pear. The results suggest that the flavor of pear is influenced not only by the content of major flavor substances but also by the collective effect of all the substances. The conclusion provides a new method and useful data for evaluating pear flavor quality, and the application of the new assessment format can improve the comprehensiveness and rationality of evaluating pear flavor quality, processing quality, and breeding based on flavor characteristics. It is of great significance for research on pear flavor quality and the development of the pear industry.

## Figures and Tables

**Figure 1 molecules-28-04794-f001:**
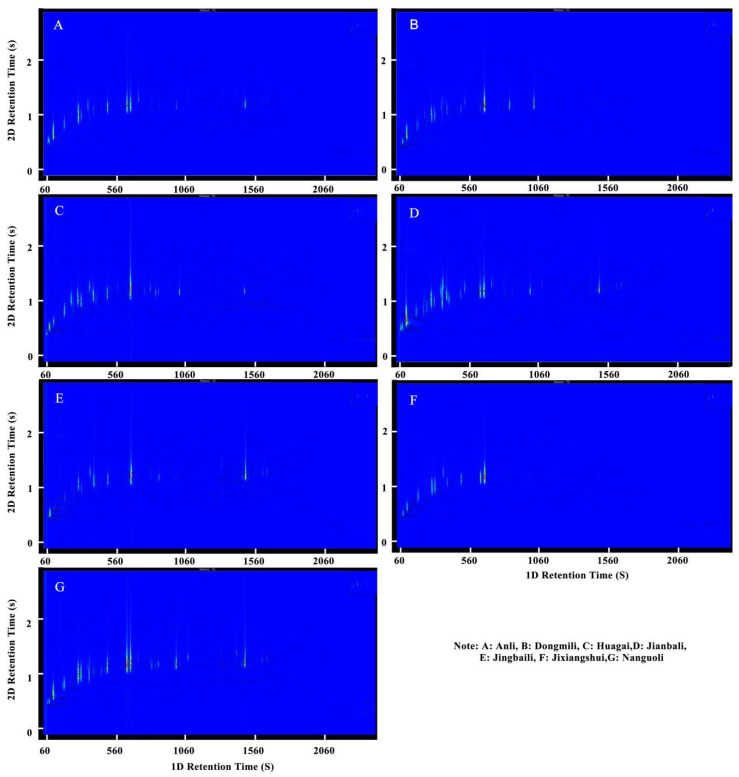
Total ion chromatograms of the aromatic components of different pear cultivars. Note: (**A**): aroma fingerprint of Anli; (**B**): aroma fingerprint of Dongmili; (**C**): aroma fingerprint of Huagai; (**D**): aroma fingerprint of Jianbali; (**E**): aroma fingerprint of Jingbaili; (**F**): aroma fingerprint of Jinxiangshui; (**G**): aroma fingerprint of Nanguoli.

**Figure 2 molecules-28-04794-f002:**
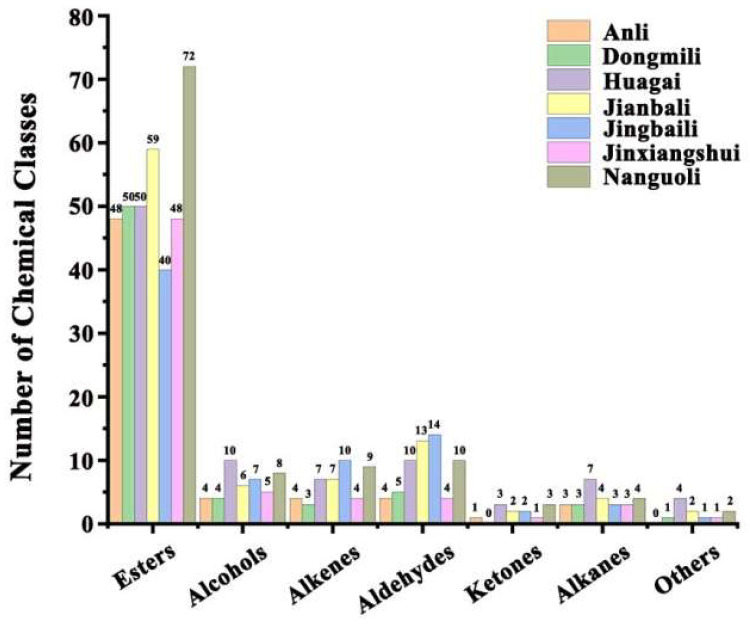
The number of different types of aroma substances in pears.

**Figure 3 molecules-28-04794-f003:**
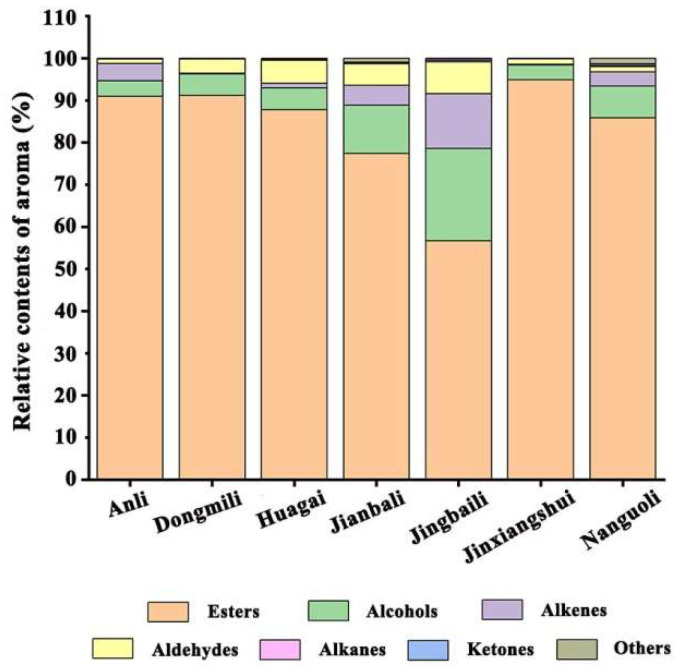
Relative content proportions of different volatile compounds in seven pear cultivars.

**Figure 4 molecules-28-04794-f004:**
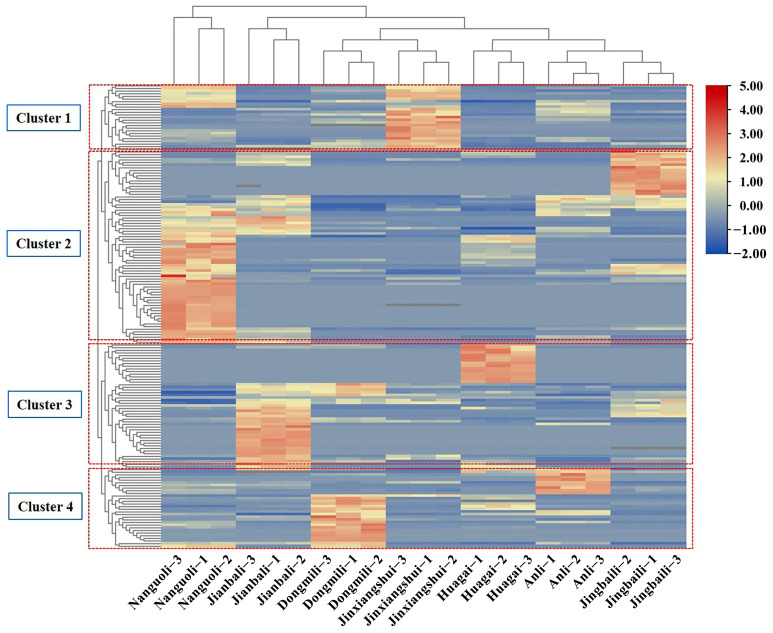
Cluster heatmap analysis of volatile organic compounds in 7 pear cultivars.

**Figure 5 molecules-28-04794-f005:**
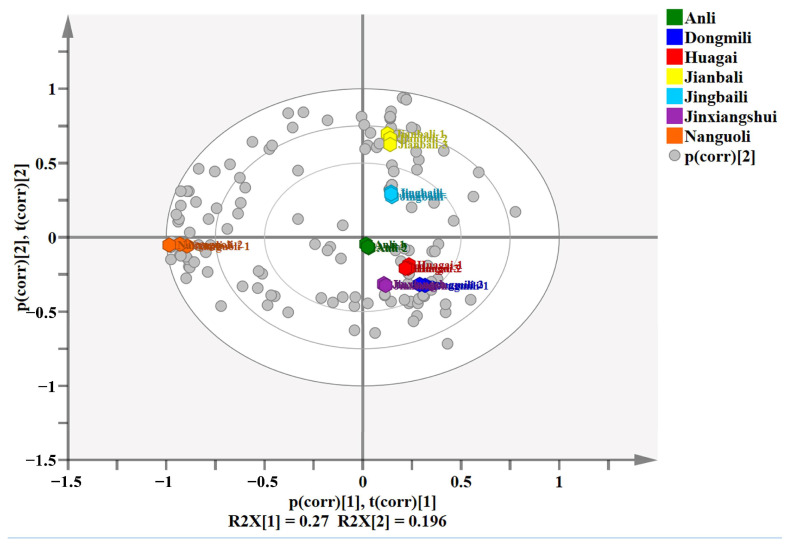
Principal component analysis of volatile organic compounds in 7 pear cultivars.

**Figure 6 molecules-28-04794-f006:**
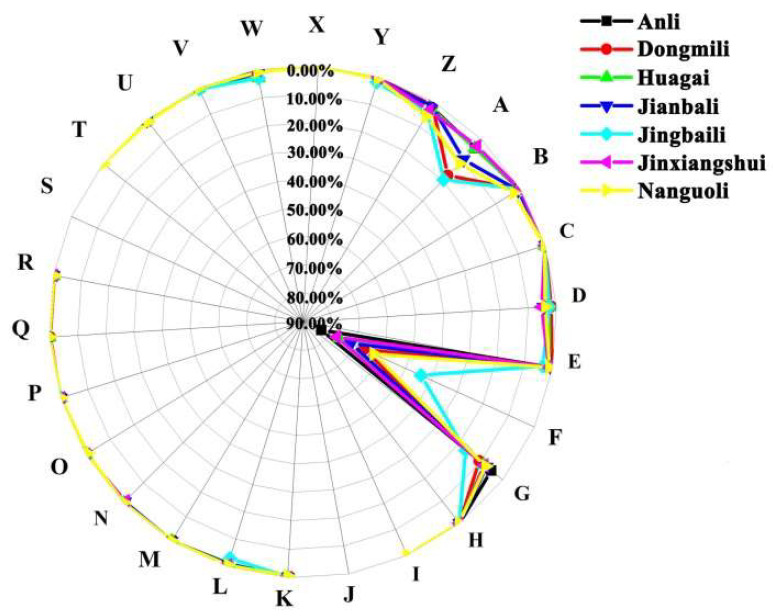
Analysis of aroma characteristic percentage distributions of 7 pear cultivars. Note: A: aliphatic; B: ice; C: citrus; D: dairy; E: edible; F: fruit; G: green; H: herb; I: iris; J: jasmin; K: konifer; L: it-chem; M: muguet; N: narcotic; O: orchid; P: phenol; Q: balsam; R: rose; S: animal; T: smoke; U: animal; W: wood; X: musk; Y: earthy; Z: solvent.

**Table 1 molecules-28-04794-t001:** Principal component variance contribution rate.

Component	Proportion (%)	Cumulative (%)	Eigenvalue
1	27.0	27.0	5.67
2	19.6	46.6	4.11
3	15.5	62.1	3.25
4	14.0	76.1	2.94
5	10.9	87.0	2.3
6	9.9	97.0	2.1

**Table 2 molecules-28-04794-t002:** The cultivars, producing areas, and sampling times of 7 pear cultivars.

Cultivars	Producing Region	Sampling Time
Anli	Huludaoshi city of Liaoning Province	27 September 2020
Dongmili	Huludaoshi city of Liaoning Province	27 September 2020
Huagai	Anshanshi city of Liaoning Province	7 October 2020
Jianbali	Huludaoshi city of Liaoning Province	15 September 2020
Jingbaili	Huludaoshi city of Liaoning Province	24 August 2020
Jinxiangshui	Huludaoshi city of Liaoning Province	12 September 2020
Nanguoli	Anshanshi city of Liaoning Province	5 October 2020

## Data Availability

All relevant data are within the manuscript and its Appendix A.

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
