# Peer review of "Exploring the Aroma Fingerprint of Various Chinese Pear Cultivars through Qualitative and Quantitative Analysis of Volatile Compounds Using HS-SPME and GC×GC-TOFMS"

_molecules, 2023, doi:10.3390/molecules28124794_

Round 1
Reviewer 1 Report
The manuscript entitled "Exploring the Aroma Fingerprint of Various Pear Cultivars through Qualitative and Quantitative Analysis of Volatile Compounds Using HS-SPME and GC×GC-TOFMS" is well performed using adequate methods for this purpose. I find this manuscript suitable for publication after some modifications:
Title:
The English name "peer" for the species is commonly used for Pyrus communis, so I suggest using the name "Chinese pear" instead. A short explanation should be presented in the manuscript to explain the main differences and also in the discussion.
Results:
- figure 5 must be modified; the resolution is generally to poor, and too much overlapping data are presented. I suggest that you use a separate chart for correlations of compounds concentration with principal components or present them in the table.
Discussion:
L453. ...were the dominant aroma components in both species... it is unclear which species?
Author Response
Response to Reviewer 1 comments: 1.Title: The English name "peer" for the species is commonly used for Pyrus communis, so I suggest using the name "Chinese pear" instead. A short explanation should be presented in the manuscript to explain the main differences and also in the discussion. Thank you. We have changed "Pear" to "Chinese Pear" in the title according to the revision suggestion. 2.Results: - figure 5 must be modified; the resolution is generally to poor, and too much overlapping data are presented. I suggest that you use a separate chart for correlations of compounds concentration with principal components or present them in the table. Thank you. It has been modified as needed. In Figure 5, display of overlapping data names are removed and used only as a demonstration of pear classification. Relevant substances in each group of pears are listed in Table S4. 3.Discussion: L453. ...were the dominant aroma components in both species... it is unclear which species? Thank you. The carbohydrate derivatives in these two kinds of pears are mainly alkenes, and also include two esters. The carbohydrate derivatives in Jingbaili are (Z,E)-α-Farnesene, (E)-β-Farnesene, α-Farnesene, (E,Z)-2,4-decadienoic acid, methyl ester, and (E,Z)-2,4-decadienoic acid, ethyl ester. Nangguoli also includes the above five substances, in addition to (Z,Z)-α-Farnesene, Limonene, β-Myrcene, (E)-β-Ocimene. At the same time, we have correct and perfected the relevant content in the manuscript.Reviewer 2 Report
This manuscript describes an in-depth investigation providing new data on the aroma composition, total aroma content, proportion and number of different aroma types, and relative quantity of each compound from various pear cultivars.
The paper answers some questions and aim to fill some of the existing gaps in the field, given that literature-derived do not include detailed information of changes in flavour composition between different types of pears.
This research provides new and useful data for production management and for improving the sensory quality of the pears. Moreover, this data aims to promote pear fruit flavour quality, deep processing and genetic improvement.
The introduction provides an overview of the current research on this topic and the motivation for this research.
The study is well designed and well written, the methodology is provided in sufficient details. The results are well illustrated and well aligned with the purpose of this paper. The results are clearly shown in Figures 1-6, proving every detail discussed in the paper.
Conclusions section must be improved according to the obtained results. Please make the conclusions more consistent, by highlighting the innovative aspects of this study that support its added value.
The references cited are relevant to this research topic, but can be supplemented. Also, the references are not written according to the journal requirements. Please edit the list of references according to the rules presented at: https://www.mdpi.com/authors/references.

Although this manuscript is well written using standard English, I suggest that the authors carefully proofread the entire manuscript to correct any grammatical or syntax errors.
Author Response
Response to Reviewer 2’s comments 1.Conclusions section must be improved according to the obtained results. Please make the conclusions more consistent, by highlighting the innovative aspects of this study that support its added value. Thank you. We have changed “This research provides a molecular-level evaluation of the flavor quality of Pyrus ussuriensis Maxim, offering theoretical support and evaluation methods for pear production management, fruit sensory quality improvement, and breeding. Moreover, this study will contribute to further research on the analysis of volatile compounds in fruits.” to “The conclusion provides a new method and useful data for evaluating pear flavor quality, and the application of new assessment format can improve the comprehensiveness and rationality of evaluating pear flavor quality, processing quality and breeding based on flavor characteristic. It is of great significance for the research of pear flavor quality and the development of pear industry.” in the manuscript according to the suggestions. 2.The references cited are relevant to this research topic, but can be supplemented. Also, the references are not written according to the journal requirements. Please edit the list of references according to the rules presented at: https://www.mdpi.com/authors/references. Thank you. The references have been revised according to the requirements of the journal. At the same time, more literature related to this research topic has been added to the references. The literature added is as follows. 1. Ndikuryayo, C.; Ndayiragije, A.; Kilasi, N.; Kusolwa, P. Breeding for Rice Aroma and Drought Tolerance: A Review. Agron. 2022, 12(7), 1726. 2. Farag, M.A.; Dokalahy, E.U.; Eissa, T.F.; Kamal I.M.; Zayed A. Chemometrics-Based Aroma Discrimination of 14 Egyptian Mango Fruits of Different Cultivars and Origins, and Their Response to Probiotics Analyzed via SPME Coupled to GC-MS. ACS Omega 2022, 7, 2377 - 2390. 3.Although this manuscript is well written using standard English, I suggest that the authors carefully proofread the entire manuscript to correct any grammatical or syntax errors. Thank you. The text has been checked once again for grammar and syntax errors, if any.Reviewer 3 Report
1. The English language in the whole manuscript requires major revision.
2. The template is not properly followed. For example, results should be before the materials and methods. The format of references in the introduction (first paragraph) is not correct.
3. The section “Research highlights” is not required in the template.
4. Figure 1 is not readable.
5. In line 177, “A total of 174 volatile organic compounds were identified in the seven pear fruits, comprising 102 esters, 14 alcohols, 20 aldehydes, 5 ketones, 18 alkenes, 8 alkanes, and 7 other substances.” The details data of identified compounds are not provided either in the manuscript or as supplementary data.
6. Statement of the seven cultivars is often wrong in the manuscript. For example, “Figure 3. Relative content proportions of different volatile compounds in seven pears.” It is not “seven pears”. It should be seven clutivars of pears.
7. The format of references doesn't match the template of molecules.
8. How the total content of aroma substances was determined? The details of qualification are not described properly in the methodology.
The English language in the whole manuscript requires major revision.
Author Response
Response to Reviewer 3’s comments 1. The English language in the whole manuscript requires major revision. Thank you. The text has been checked once again for grammar and syntax errors, if any. 2. The template is not properly followed. For example, results should be before the materials and methods. The format of references in the introduction (first paragraph) is not correct. Thank you. We have put the results before the materials and methods as template required and modified the quotation format in the introduction and the full manuscript. 3. The section “Research highlights” is not required in the template. Thank you. The "Research Highlights" section has been deleted in accordance with the modification requirements. 4. Figure 1 is not readable. Thank you. Figure 1 has been reuploaded, and the text containing Figure 1 has been saved to PDF format for uploading. 5. In line 177, “A total of 174 volatile organic compounds were identified in the seven pear fruits, comprising 102 esters, 14 alcohols, 20 aldehydes, 5 ketones, 18 alkenes, 8 alkanes, and 7 other substances.” The details data of identified compounds are not provided either in the manuscript or as supplementary data. Thank you. Supplementary files have been uploaded again, and Table S1 lists all the above data. 6. Statement of the seven cultivars is often wrong in the manuscript. For example, “Figure 3. Relative content proportions of different volatile compounds in seven pears.” It is not “seven pears”. It should be seven clutivars of pears. Thank you. The expressions of the seven varieties in the text have been modified, and the captions in Fig. 3 have been modified to “Fig. 3. Relative content proportions of different volatile compounds in seven pear cultivars.” 7. The format of references doesn't match the template of molecules. Thank you. All references have been corrected according to the template format as needed. 8. How the total content of aroma substances was determined? The details of qualification are not described properly in the methodology. Thank you. The total aroma substance content of every pear is the sum of the relative contents of all aromas identified in each fruit. The details of identification are further described in section 3.2.5 according to the revision suggestion.Round 2
Reviewer 3 Report
Figure 1 is still not readable.
It will be nice to ask a native speaker to modify the language.
Author Response
Responses to Reviewer 3’s comments: 1. Figure 1 is still not readable. Thank you. Image 1 in Tiff format has been converted to JPG format in manuscript. Because there is only one upload access, but I still need to upload the revised manuscript, so I uploaded the revised manuscript. At the same time, I send Figure 1 to editor and ask her to forward a TIFF, JPG and PDF format picture to you. I don't know why the Figure 1 was still not readable, So I sent pictures in different formats. 2. It will be nice to ask a native speaker to modify the language. Thank you. The English of the manuscript was further modified.